# Design and Construction of pH-Selective Self-Lytic Liposome System

**Ayumi Kashiwada *, Kana Namiki and Haruka Mori**

Department of Applied Molecular Chemistry, College of Industrial Technology, Nihon University,
1-2-1 Izumi-cho, Narashino, Chiba 275-8575, Japan; cika14137@g.nihon-u.ac.jp (K.N.);
jcha16040@g.nihon-u.ac.jp (H.M.)

* Correspondence: kashiwada.ayumi@nihon-u.ac.jp; Tel.: +81-47-474-2564

**Abstract:** Liposomes are well-investigated drug or gene delivery vehicles for chemotherapy, used by taking advantage of their biocompatibility and biodegradability. A central question on the construction of intracellular liposomal delivery systems is to entrap the liposomes of interest in the highly acidic and proteolytic endosomal environment. In the other words, it is essential that the liposomes release a therapeutic drug into the cytosol before they are degraded in the endosome. As a strategy to enhance the endosome escape, the self-lytic liposomes with acidic pH-selective membrane active polypeptide are considered highly effective. Here, an acidic pH-selective membrane-lytic polypeptide (LPE) and its retro isomer (rLPE) were designed, and then their membrane-lytic activities against EggPC liposomes were determined. It was noticed that the rLPE polypeptide showed an increase in activity compared with the LPE polypeptide. Furthermore, the rLPE polypeptide was conjugated to liposomes via a flexible Gly-Gly-Gly-Gly linker to facilitate the pH-selective content release. The rLPE anchoring liposomes exhibited distinctly different contents release behavior at physiological and endosomal pHs, namely typical contents release from liposomes was positively observed at acidic pH range. The overarching goal of this paper is to develop efficient pH-selective therapeutic delivery systems by using our findings.

**Keywords:** pH-selective liposomes; membrane lytic polypeptides; acidic pH; endosome escape

## 1. Introduction

Liposomes are artificial spherical vesicles with a bilayered membrane structure formed with the self-assembly of nontoxic phospholipids in aqueous solution [1–3]. Their lipid bilayer of liposomal vesicles enables their efficacy as drug delivery carriers due to their versatile nature in encapsulating both hydrophobic and hydrophilic drugs within nonpolar bilayer regions of the membrane and the aqueous core, respectively [4]. Furthermore, it is well known that liposomes with a diameter around 100 nm are able to accumulate in tumorous cells via the enhanced permeability retention (EPR) effect [5,6]. After reaching target cells, the contents release from liposomes is generally achieved through a passive mechanism. [7]. Thus, demand for the construction of liposome-based systems that include mechanisms to facilitate suitable stimuli-responsive contents release can be expected to increase.

Among the stimuli, the drop in pH is the most typical stimulus due to the pH gradients found in the physiological pathways. For instance, tumor cells are slightly more acidic than normal cells [8,9]. Even within cell compartments there is a pH value gradient. Cytosol has a physiological pH value around pH 7.4, whereas the endosome features a pH between 6.5 to 5.0 [10]. The release of the encapsulated contents from liposomes in response to light and temperature needs an additional outer device for their activation [11–14]. On the other hand, the pH-selective system needs no additional

outer device. Therefore, the pH-selective (or pH-responsive) liposomes can serve as attractive carriers in a drug delivery system.

The entrapped contents release from liposomes is provided by the disturbance of the lipid membrane induced by pH responsive molecules. Among the suitable candidates for membrane disturbing agents, polypeptides have received considerable attention. Different kinds of polypeptides have been proposed, including coiled coils and membrane lytic polypeptides [15–18].

We have designed a melittin mimetic lipid membrane disturbing polypeptide (LP) and demonstrated lytic activity toward the biocompatible EggPC liposomes; when interacting with surfaces of the lipid bilayers, it forms a hydrophobic helix in the lipid membrane in a wide pH range. Moreover, we have proposed LP analogues with endosomal acidic pH selective lytic activity [19].

In this study, we have designed a pH-selective self-lytic liposome system by using a melittin-inspired polypeptide device. Although there have been some reports about the design of pH-selective self-lytic liposomes [20,21], no research has yet been carried out to construct a membrane lytic polypeptide-based system that responds so sensitively to a change in pH (from 7.4 to 5.0). Here we report the design of novel melittin-inspired membrane lytic polypeptides in order to use the pH-selective segment for a membrane penetrating device. In addition, we report herein the preparation and characterization of a pH-selective liposome system combined with a polypeptide device. This disclosed pH-sensitive liposome system by the use of the artificial polypeptide device is unique and different from the conventional stimuli-responsive system incorporating a sensitizer such as switchable surfactant. Our results shed light onto the polypeptide conjugated liposomes as drug delivery system (DDS) vehicles and provide some fundamental insight into the use of the membrane lytic polypeptides for therapeutic applications.

## 2. Materials and Methods

### 2.1. Synthesis and Purification of the Membrane Lytic Polypeptides

All polypeptides and polypeptide conjugate were synthesized on Rink amide resin. Polypeptide synthesis was performed by the standard Fmoc protocols using HOBt/HBTU activation. The deprotection and cleavage from the resin were carried out by the treatment with an 89:10:1 mixture (2 mL) of trifluoroacetic acid (TFA), triisopropylsilane, and water for 2 h. Crude polypeptides were purified by the RP-HPLC on a HITACHI D-7000 with an Inertsil ODS-3 column (10 μm, 250 × 10 mm, GL Sciences Inc., Tokyo, Japan) using a linear gradient of solvent A in solvent B over 30 min (solvent A, 0.1% TFA in water; solvent B, 0.1% TFA in acetonitrile). The purity of the polypeptides were confirmed by analytical HPLC (Inertsil ODS-3 column, 5 μm, 250 × 4.6 mm, GL-science, Tokyo, Japan). Molecular weight of the purified polypeptides was identified by a high resolution ESI-TOF MS by using an Agilent 6210 ESI-TOF LC-MS spectrometer (Agilent Technologies Inc., Sanat Clara, CA, USA).

### 2.2. Preparation of Liposomes

Small unilamellar liposomes were prepared by 1.3 mg of EggPC dissolved in chloroform, followed by drying in a round-bottom flask in vacuo. The dried thin film of EggPC was hydrated in a 1 mL of 100 mM phosphate buffer solutions (containing 100 mM sodium chloride) with different pH values. The resultant the multilamellar vesicle was subjected to 5 freeze-thaw cycles. Then, the solution of heterogeneous size of unilamellar liposomes is passed through a polycarbonate filter with 100 nm diameter pores (Merck, Darmstadt, Germany) a total of 21 times on Mini-Extruder Set (Avanti Polar Lipids, Inc., Alabaster, AL, USA).

Liposomes consisting of 2.0 mM EggPC and incorporated with varying amounts (0.5, 1.0, 1.5 and 2.5 mol%) of the rLPE-St were also prepared by the same procedure described above.

### 2.3. Calcein Leakage Assay

Membrane activity of the designed polypeptides was characterized by monitoring the increase of fluorescence intensity upon leakage of the self-quenched calcein (75 mM) to the surrounding medium of the liposomes. Then, the EggPC liposomes (2.0 mM) were mixed and incubated with polypeptide samples. The fluorescence of the released calcein was measured using a HITACHI F-2500 fluorescence spectrometer at a 488 nm/520 nm excitation/emission wavelengths. The maximum release (100%) was achieved by the addition of Triton X-100 (0.5% final concentration).

For investigating a detailed pH dependence of the rLPE-St anchoring liposomes, 0.1 mL of the calcein (75 mM) contained liposome solution (pH 10.0) was added to a 2.9 mL of 100 mM phosphate buffer at the desired pH (8.5, 8.0, 7.5, 7.4, 7.0, 6.5, 6.0, 5.5, 5.0). The fluorescence of the released calcein was measured using a HITACHI F-2500 fluorescence spectrometer at a 488/520 nm excitation/emission wavelengths. The maximum release (100%) was achieved by the addition of Triton X-100 (0.5% final concentration).

The percentage of calcein leakage was calculated as follows: calcein leakage $\% = (F - F_0)/(F_t - F_0) \times 100$, in which $F$ and $F_t$ represent the fluorescence intensity of calcein before and after the addition of Triton X-100, respectively, and $F_0$ represents is the initial fluorescence before the addition of lytic polypeptides.

### 2.4. Liposome-Accessibility Assay

The EggPC liposomes incorporated 0.5% 7-nitro-2,1,3- benzoxadiazole (NBD)-labeled phosphatidylethanolamine (NBD-PE) were prepared by the same procedure as described above. The NBD-PE of the outer lipid layer, which were exposed to the external buffer solution, were reduced and quenched by dithionite ions. Thus, only the inner lipid layer of the liposomes contributed to the emission of NBD and the fluorescence was measured with a HITACHI F-2500 fluorescence spectrometer at a 475/530 nm excitation/emission wavelengths. Reduction of the outer lipid layer of NBD-PE incorporated liposomes was carried out as follows: The solutions of NBD-PE incorporated liposomes (2.0 mM) and 10 mM sodium dithionite (in 100 mM phosphate buffer containing 100 mM sodium chloride) were mixed at 25 °C, then incubated for 60 min. The reduction of NBD in the outer layer was confirmed by the decrease of fluorescence at 530 nm. After the reduction of the outer layer of liposomes, free sodium dithionite was removed by gel filtration chromatography on Sephadex G-25. Then the liposome was concentrated and adjusted to 2.0 mM by using a Centricon Plus-70 (Membrane NMWL: 3 kDa). To estimate the accessibility to the internal aqueous compartment of the lipid vesicles, we assessed the reduction of NBD in the inner monolayer. Fluorescence spectra of NBD were measured following the addition of 2.9 mL of fresh sodium dithionite solution (100 μM) at the desired pH, and then 0.1 mL of the liposome solution (pH 10.0) was added. The liposome accessibility was evaluated by the quenching of NBD fluorescence in the inner bilayer with bilayer permeated reductant for 40 min.

### 2.5. CD Measurements

CD data were collected on a Jasco J-820 CD spectropolarimeter with a PTC-348WI peltier thermostat at 25 °C. Sample solutions were prepared in phosphate buffer solutions containing 100 mM sodium chloride. Concentrations of EggPC anchoring the rLPE-St was the same as other experiments (2.0 mM) when CD measurements were carried out. Reported CD data from 260 to 200 nm are averaged over 50 scans.

### 2.6. Dynamic Light Scattering (DLS) Measurements

DLS measurements of the EggPC liposomes anchoring the rLPE-St were characterized on an N5 Plus (Beckman Coulter Inc., Brea, CA, USA) with a He-Ne linearly polarized laser operating at 632.8 nm. All the measurements were carried out at a scattering angle of 90.0°. Size distributions were analyzed and calculated with a CONTIN algorithm.

*2.7. Zeta Potential Measurements*

Zeta potentials were determined from the electrophoretic mobility, measured using an ELSZ-2000Z (Otsuka Electronics Co. Ltd., Hirakata, Osaka, Japan). The zeta potentials were averaged over 3 measurements in each sample.

## 3. Results and Discussion

*3.1. Design of Melittin-Mimetic Lytic Polypeptides and Their Retro-Analogs*

In order to design artificial membrane-lytic polypeptides, we have used melittin with a strong membrane destabilizing activity as a template. An artificial lytic polypeptide (LP) has a de novo designed helix-promoting hydrophobic Leu-Ala segment and a C-terminus cationic residues from melittin (Figure 1). By the use of the LP, we have evaluated the membrane destabilizing activity against EggPC liposomes. Moreover, we have constructed LP analogues with membrane lytic activity at the endosomal condition. One of the polypeptides, the LPE which was altered, had a hydrophobic helical segment of the LP which showed effective pH-dependent lytic activity (Figure 1).

**Figure 1.** Amino acid sequences of melittin-inspired membrane lytic polypeptides and their retro isomer polypeptides.

An effective approach for the construction of artificial membrane-lytic polypeptides is to design retro analogs against native polypeptides. Retro analogs have the same net charge and hydrophobicity, though the direction of hydrophobic moment and the cationic topology are different from native polypeptides. There has been reported that the reverse sequence of a small synthetic polypeptide with the alternating sequence of Lys and Trp enhanced antimicrobial activity [22]. In addition, retro analog of melittin and its stearoyl derivative have been reported to show melittin-like lytic activity and the decreased toxicity [23,24]. The hydrophobic segment of LP and LPE polypeptides form helical structure in case of inserting into lipid bilayers. Due to specific spatial orientation of dipoles, helix moieties may be characterized by their macrodipole which is a resultant of summation of dipoles of individual polypeptide bonds. Thus, characterization of the retro-analogs is important to understand better the structure–function relationships that exist in the melittin-mimetic polypeptides.

Here, we examine the structural consequences of membrane lytic activities of LP and LPE polypeptides, considering its reversed (retro) sequence (Figure 1).

*3.2. Membrane Lytic Properties of the Designed Polypeptides*

The acidic pH-selective membrane lytic activity of retro polypeptide analogs were investigated by a calcein-leakage assay with liposomes consist of EggPC [25]. The fluorescent dye calcein loaded into the liposomes above the self-quenching concentration is released to the medium outside the liposomes

when membrane disruption occurs. Consequently, the self-quenching of calcein would be relieved and the fluorescence of the sample would recover.

On addition of the 20 μM of rLP polypeptide to EggPC liposomes at pH 7.4 and 25 °C, 60% release of the entrapped calcein was observed over the first 20 min after which the rate of release decays significantly until the release of a constant amount of the dye is attained (Figure 2A). The membrane lytic activity against EggPC liposomes indicated that the retro analog rLP showed an increase of activity, compared to that of the normal LP polypeptide. Similar calcein-release behavior against the rLP addition was also found at pH 5.0 (Figure 2A). There has been much discussion that the membrane lytic or antimicrobial activity depends on the polypeptide sequence. Some reports have indicated that the retro analogs of membrane lytic polypeptides possess higher activity than the normal polypeptides [22–24]. Therefore, the increased membrane lytic activity of rLP against EggPC liposomes is not inconsistent with the previous reports.

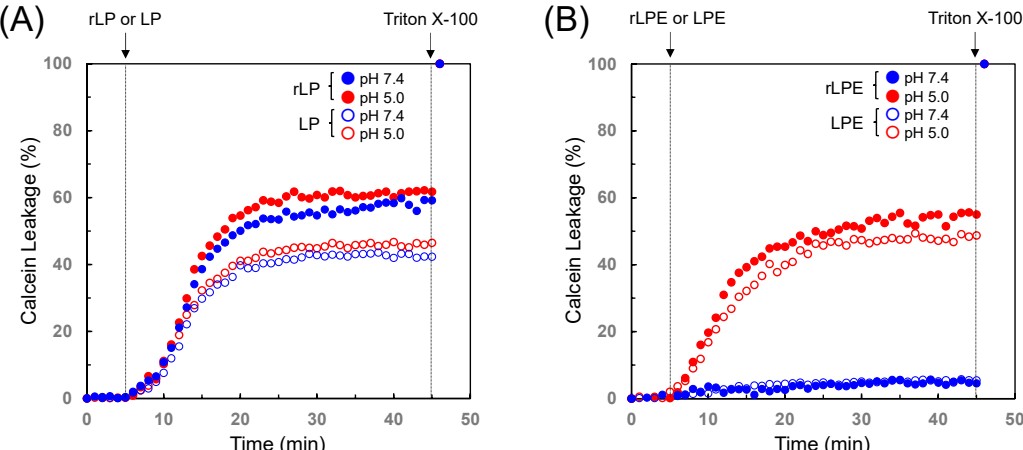

**Figure 2.** Membrane lytic activities of the LP and rLP polypeptides (**A**) and the LPE and rLPE polypeptides (**B**). A typical calcein-leakage assay with the normal- and the retro-polypeptides was carried out at both physiological (pH 7.4) and endosomal (pH 5.0) conditions. Calcein leakage from EggPC liposomes was measured every 0.5 or 1 min a following the addition of 20 μM of polypeptide at 25 °C.

In order to investigate the lytic activity of the pH-selective polypeptides by using the calcein-leakage assay, the changes in the behavior of calcein leakage induced by the LPE polypeptide or its retro analog (rLPE) were characterized. As shown in Figure 2B, the slightly higher acidic pH-selective lytic activity was observed in addition of the retro sequence analog rLPE to EggPC liposomes. In the end, the rLPE polypeptide induced 56.0% release of the encapsulated calcein while 48.8% release was observed by the addition of the LPE polypeptide.

### 3.3. Design of Acidic-pH-Selective Membrane Lytic Polypeptide Conjugate

The LP polypeptides have an ability to interact with liposomal surfaces and to form a hydrophobic helix in the lipid membrane at acidic and neutral pHs. On the other hand, one of the LP analogues, LPE showed membrane lytic properties only at endosomal acidic pH [19].

In this study, pH-selective lytic polypeptide LPE and the retro analog (rLPE) were synthesized and compared with membrane permeability. In the results reported above, the retro analog, rLPE showed a high extent of calcein release from EggPC liposomes compared to that of the normal LPE polypeptide. On the basis of this results, the sequence, rLPE, was used as the pH-selective membrane lytic segment held in lipid bilayers.

The conjugate based on the rLPE polypeptide (rLPE-St; (Figure 3)) was synthesized according to the standard Fmoc polypeptide synthesis methods. To anchor this conjugate into the liposomal membranes, stearoyl group was coupled to the N-terminus of the rLPE segment via a flexible Gly-Gly-Gly-Gly linker.

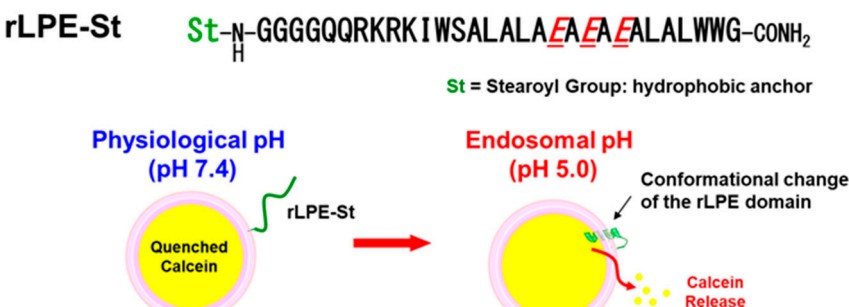

**Figure 3.** Schematic illustration of the pH-selective self-lytic liposome system by the use of the acidic-pH-selective membrane lytic polypeptide conjugate (rLPE-St).

An illustration of our designed pH-selective self-lytic liposome system is also shown in Figure 3. The rLPE polypeptide domain expects to exist in a flexible water-soluble structure at physiological conditions due to the deprotonation of the glutamate residues. Under these conditions, it does not lead to contents (calcein) release from liposome because of the lower membrane lytic activity of the rLPE domain. The rLPE domain, however, does protonate of glutamate and undergo a conformational change to a hydrophobic α-helical structure at reducing the pH of the surround solutions to an endosomal acidic condition. This pH-selective conformational change allows to promote membrane lytic activity of the rLPE domain. The pH-dependent electrostatic properties of the surface of self-lytic liposomes would be close relation to lytic activities. Therefore, we have gained an insight into a character on surfaces of the 1.0 mol% of rLPE-St anchoring liposomes from zeta potential measurements carried out at pHs 7.4 and 5.0. The results of zeta potential measurements are summarized in Table S1. The zeta potential values of the rLPE-St anchoring liposomes were critically dependent on the pH. The surfaces of vesicles had weakly negative (−1.28 mV) at pH 7.4, while increased positive charge (+1.78 mV) at pH 5.0, lower than the pKa value of the glutamate side chain. This results of zeta-potential measurements support that reduction of the electrostatic repulsion and hydrophobic membrane permeable helix formation of rLPE segment at weakly acidic pH permits contents release from liposome.

### 3.4. pH-Selective Lytic Properties of the rLPE-St Anchoring Liposomes

The effects on calcein release from liposomes anchoring the 1.0 mol% of rLPE-St were investigated at both endosomal and physiological conditions. The rate of efflux was monitored over 40 min as no appreciable increase was observed after this interval. For the rLPE-St anchoring liposomes at pH 5.0, it was observed that the majority of release carried out for the first 15 min, after which the rate of release decays significantly until the release of a constant amount of the dye is attained (Figure 4). In contrast, significant efflux of calcein was not observed from the EggPC liposomes at pH 7.4 (Figure 4).

The effective release (40% to 50% release) of calcein from liposomes was achieved above 1.0 mol% of rLPE-St at pH 5.0 (Figure 5 and Figure S2). However, higher amount of rLPE-St leads to slight increase of calcein release at unsuitable physiological pH (Figure 5 and Figure S2). Taking the above results into account, we have estimated that the 1.0 mol% of rLPE-St anchoring liposomes shows suitable endosomal pH selective calcein release behavior.

We also examined the dependence of a variety of pH on the calcein release from rLPE-St anchoring liposoms. The total percentage of calcein released from 1.0 mol% of rLPE-St anchoring liposomes was measured after 40 min at various pH. Significant calcein release (c.a. 12% release) was observed to occur at pH 6.5. With the further decrease of the pH at the surrounding buffer solution, the total percentage of calcein releases reached to a maximum of c.a. 48% at pH 5.0 (Figure 6 and Figure S3).

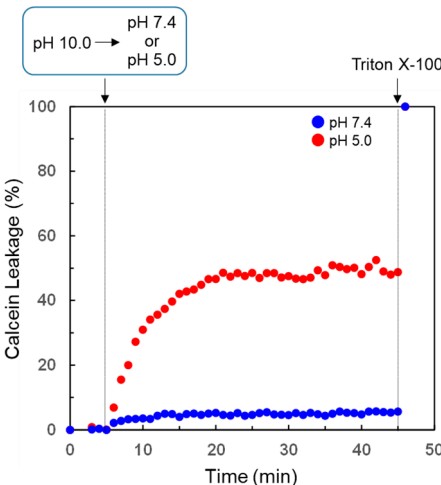

**Figure 4.** pH-selective calcein release behavior from liposomes anchoring the 1.0 mol% of rLPE-St. The calcein-entrapped liposomes were prepared at pH 10.0. After 5 min incubation, the pH was lowered to a value of 7.4 or 5.0.

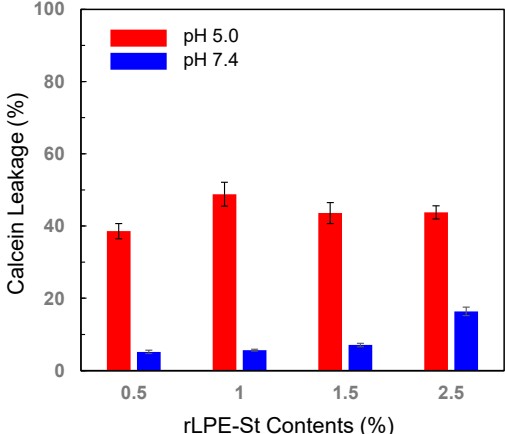

**Figure 5.** Percent calcein release over 40 min from 0.5, 1.0, 1.5 and 2.0 mol% rLPE-St anchoring liposomes triggered with switching the pH from 10.0 to 5.0 or 7.4. Error bars: SD of data from 5 measurements.

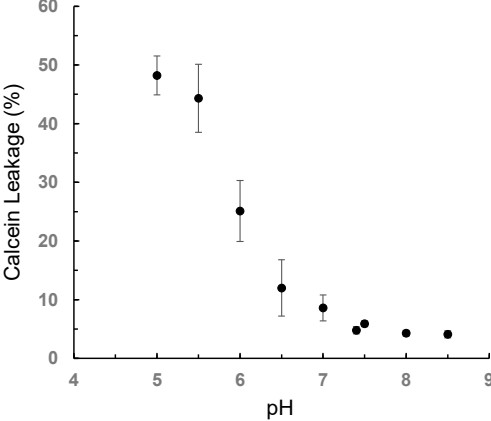

**Figure 6.** pH-dependence of the total amount of calcein release from liposomes anchoring the 1.0 mol% of rLPE-St. Error bars: SD of data from 5 measurements.

It has been well established that mellitin and its model polypeptides are able to be inserted into phospholipid bilayers [23,26–28]. However, the pH-selective mechanisms of impairing rLPE-St anchoring liposomal bilayers remain elusive. To assess the processes of the polypeptide mediated

membrane turbulence and the accessibility, we monitored the quenching of NBD on the outer layer of NBD-PE labeled liposomes by reaction with sodium dithionite in the external solution. As the liposomal membrane is not permeable to the dithionite anion, a specific quenching of the NBD in the outer lipid layer can be observed [29]. Immediately after the addition of sodium dithionite to the rLPE-St anchoring NBD-labeled liposomes at pH 10.0, the fluorescence intensity reduced to about 45% of its initial value. This results indicate that the reductant molecules affect the readily reactive outer lipid layer of the liposomes. [30]. Further quenching of NBD at the inner lipid layer by the reaction with the dithionite anion could be observed when the pH at the surrounding medium was decreased to pH 5.0 (Figure 7). In contrast, the rLPE-St anchoring NBD-labeled liposomes showed almost no particular quenching at physiological pH. This pH-selective dithionite influx property is good agreement with the calcein release behavior of rLPE-St anchoring liposoms.

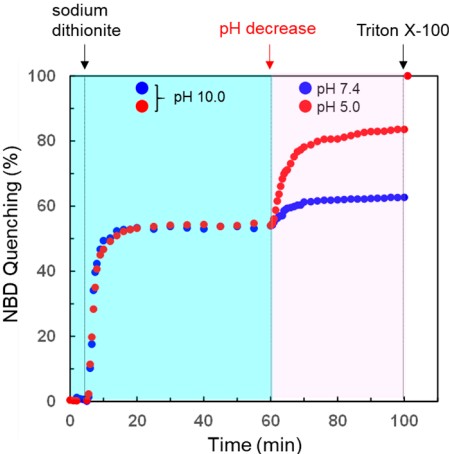

**Figure 7.** Membrane lytic activity of the liposomes anchoring the 1.0 mol% of rLPE-St characterized by the membrane accessibility assay. Membrane accessibility was estimated from quenching of NBD dye-labeled lipids by using a membrane-impermeable reductant sodium dithionite. Measurements were carried out at 25 °C.

The variation of the liposome size in the calcein release process was followed by dynamic light scattering (DLS). For the rLPE-St anchoring liposomes (total lipid concentration: 2.0 mM) at pH 5.0, the sudden increase of the mean diameter due to the turbulence of vesicle surfaces by the contact of the polypeptide segment was observed, and then the size of liposomes were decreased and returned to the original diameter (around 110 nm). Similar size change of liposomes due to the turbulence of vesicle surfaces was also observed in liposomal binding and fusion systems [31]. After this membrane turbulence phenomena, a significant change in the diameter was not observed, indicating that vesicle fusions or aggregations did not occur (Figure S4). This finding demonstrated that endosomal-pH selective membrane lytic conjugates rLPE-St, after selective interaction to the own liposomal vesicle, inserts into lipid bilayers, and leads to contents leakage from liposomes.

*3.5. Secondary Structure of the rLPE Segment Anchored into the Liposomal Membranes*

Antimicrobial polypeptides with α-helical structure show superior antimicrobial activities that facilitate the polypeptides to be bound to the surface lipids and lead to the permeation or disruption of the bilayered membrane [26,32]. Moreover, the artificial pH-selective membrane lytic polypeptide, LPE also assumes a helical structure at pH 5.0 [19].

Here, to estimate the pH-selective conformational change of the rLPE-St polypeptide anchoring into the liposome bilayers, circular dichroism (CD) analysis were applied at both physiological and endosomal pHs.

The 1.0 mol% of rLPE-St polypeptide anchored into the liposomal membranes adopting a predominantly helical structure at an endosomal pH, which possessed characteristic signals at

208 and 222 nm. On the other hand, this polypeptide existed a random coil at pH 7.4 from the resultant CD signal (Figure 8). This result revealed that the hydrophobic segment of the rLPE-St polypeptide showed obvious change in the secondary structure at endosomal acidic conditions. This result is also in good agreement with the pH-selective calcein-release behavior of the rLPE-St anchoring liposome. The dependence of rLPE-St composition upon the CD spectra was also characterized, and 2.5 mol% of rLPE-St anchoring liposome showed the increase in negative ellipticity at 222 nm even at pH 7.4 (Figure S5). This unexpected appearance of helicity at physiological pH would be closely correlate to unsuitable calcein-release behavior at physiological pH shown at Figure 5.

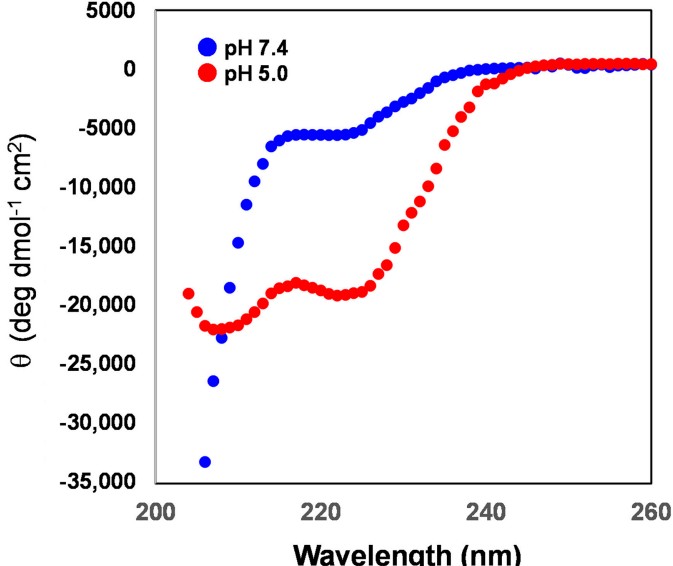

**Figure 8.** Circular dichroism (CD) spectra of the rLPE-St (1.0 mol%) anchored into the liposomal membranes at pH 7.4 and pH 5.0. Measurements were carried out at 25 °C.

## 4. Conclusions

In this paper, we presented the design and characterization of the melittin mimetic pH-selective lytic polypeptide LPE and its retro isomer (rLPE). Gratifyingly, both the LPE and the rLPE polypeptides exhibited significantly pH selective membrane lytic activity against EggPC liposomes. Especially, the rLPE polypeptide showed slightly higher membrane lytic activity judging from calcein-leakage assay. From this knowledge, we also designed the rLPE polypeptide and stearic acid conjugate (rLPE-St) that operate under weakly acidic conditions as a liposomal membrane lytic device. The rLPE-St anchoring liposomes were developed, can be activated by an acidic condition to the hydrophobic segment of the rLPE polypeptide, and, in the process, incorporate into the bilayers, supporting liposomal contents release. Our results provide useful insight onto the ongoing efforts and design considerations of unique and effective self-lytic liposomes with a pH-selectivity for the development of efficient therapeutic drug or gene delivery systems.

**Supplementary Materials:** The following are available online at http://www.mdpi.com/2227-9717/8/12/1526/s1, Figure S1: Analytical HPLC chromatograms and mass spectra of the synthetic membrane-lytic polypeptides and the designed acidic-pH-selective membrane lytic polypeptide conjugate (rLPE-St). Figure S2: pH-dependent calcein release behavior from liposomes anchoring the various amounts (0.5 mol% to 2.5 mol%) of rLPE-St. The calcein-entrapped liposomes were prepared at pH 10.0. After 5 minutes incubation, the pH was lowered to a value of 7.4 or 5.0. Figure S3: pH-dependent calcein release behavior from liposomes anchoring the 1.0 mol% of rLPE-St. The calcein-entrapped liposomes were prepared at pH 10.0. After 5 minutes incubation, the pH was lowered to a target pH. Figure S4: Mean diameter of the liposomes anchoring the 1.0 mol% of rLPE-St from DLS measurements during pH-activated membrane lytic process. Figure S5: Circular dichroism (CD) spectra of the rLPE-St (0.5, 1.5, and 2.5 mol%) anchored into the liposomal membranes at pH 7.4 and pH 5.0. Measurements were carried out at 25 °C. Table S1: Zeta potentials of the rLPE-St anchoring liposomes.

**Author Contributions:** Conceptualization, A.K.; methodology, A.K., K.N. and H.M.; formal analysis and investigation, K.N. and H.M.; writing—original draft preparation, A.K.; writing—review and editing, A.K.; supervision, A.K. All authors have read and agreed to the published version of the manuscript.

**Funding:** This research was funded by Nihon University Multidisciplinary Research Grant for 2019 and 2020.

**Conflicts of Interest:** The authors declare no conflict of interest.

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
