# Peer review of "Design and Construction of pH-Selective Self-Lytic Liposome System"

_processes, doi:10.3390/pr8121526_

Round 1
Reviewer 1 Report
Remarks about the manuscript:
In this manuscript, the authors have presented the design and construction of a pH-selective self-lytic liposome system. It is an interesting topic (passive drug delivery), which shows a liposomes lysis by the incorporated peptide in the response to the surrounding pH on endosomes and subsequent cargo delivery, which is very interesting. The data presented in this manuscript is really good with a proper explanation about synthesis and description methodology.
In my view, this manuscript (ID: processes-993968) can be accepted after considering the following comments:
1- Authors should improve the introduction section, and compare different studies and highlight the advantages of their study.
2- Cross-check whether liposomes extruded with odd numbers (page 2, line 84). The most review mentioned 11, 13, or 15 but not 10).
3- The sequence of amino acid in figure one is not so clear, can be improved by increase font size or remove boldness.
4- Figure symbols should be LPE and rLPE for figure 2(B).
5- Authors claim that 1.0 mol% of rLPE-St is best for membrane lysis, Authors should explain in the main text, why 1.5 and 2.0 mol% increase for 7.5 pH and decrease for 5.0 pH compares to 1.0 mol%. May analyze these effects with DLS data and CD measurement.
6- Authors should carefully check their text because there are some repetitions of words in the text (i.e. page 6, line 218).
7- Authors should try to check these peptide incorporated liposome’s zeta-potential and cytotoxicity as they mentioned these liposomes for nanomedicine applications.
Author Response
We wish to express our appreciation to the Reviewer for his or her insightful comments, which have helped us significantly improve the paper.
1- Authors should improve the introduction section, and compare different studies and highlight the advantages of their study.
→ We have revised the summary of the introduction section (p. 2, lines 64-68) to reflect the advantages of our work.
2- Cross-check whether liposomes extruded with odd numbers (page 2, line 84). The most review mentioned 11, 13, or 15 but not 10).
→ We have rewritten the number of extrusion “21 times (= 10 reciprocations + 1)” (p. 2, line 89)
3- The sequence of amino acid in figure one is not so clear, can be improved by increase font size or remove boldness.
→ We have reflected your comment and prepared new Figure 1.
4- Figure symbols should be LPE and rLPE for figure 2(B).
→ We have reflected your comment and prepared new Figure 2(B).
5- Authors claim that 1.0 mol% of rLPE-St is best for membrane lysis, Authors should explain in the main text, why 1.5 and 2.0 mol% increase for 7.5 pH and decrease for 5.0 pH compares to 1.0 mol%. May analyze these effects with DLS data and CD measurement.
→ We have characterized the structure of rLPE domain by CD measurements for 0.5, 1.5 and 2.0 mol% rLPE-St anchoring liposomes (Figure S5). Moreover, with respect to 2.0 mol% rLPE-St anchoring liposome, the relationship between unsuitable calcein-release behavior and secondary structure of rLPE was explained in the revised text (p. 8, lines 314-318)
6- Authors should carefully check their text because there are some repetitions of words in the text (i.e. page 6, line 218).
→ This error has been corrected in accordance with the reviewer's comment.
7- Authors should try to check these peptide incorporated liposome’s zeta-potential and cytotoxicity as they mentioned these liposomes for nanomedicine applications.
→ We have characterized the pH-dependent electrostatic properties of the surface of self-lytic liposomes from zeta-potential measurements for 1.0 mol% rLPE-St anchoring liposomes (Table S1). Moreover, we have revised the text (p. 6, lines 225-234) to explain the result of zeta-potential measurements.
Reviewer 2 Report
Dear Authors,
The submitted manuscript is dealing with self-lytic liposomes applying melittin-inspired peptides. The general approach using melittin inspired peptides is well recognized within the last decade, however deeper knowledge of the fundamental mechanism of such liposomes, but also concerns for potential application are of great scientific interest.
Detailed recommendations:
78-86: Liposomes consisting of EggPC are prepared by conventional, small scale technique. Although this composition is applicable, it is not explained why only this composition was selected, especially for potential human application.
150-156: The design of retro analogs, with all their known limitations, should be explained in more detail, explicitly the design of the melittin analogs and the comparison to still published .analogs It would significantly improve the quality of this paper, when not only references are cited, but also the models and the different results would be discussed.
163-182: the calcein release assay was performed to compare the different lytic activities of the added peptide variants in solution. In both figures (2A and 2B) the same legend is used
Furthermore, figure 2B and the corresponding text is not comprehensible and need to be thoroughly checked.
227-231:
The statement of proportional calcein release is not obvious. In addition, the legend of the x-axis of figure 5 needs to be revised
Figure 6: X-axis is not marked.
264-270:
Size variation is seen under both conditions (S4), but more dominant at pH 5.0. Maybe, supporting influences caused this behavior. Comparative, fluid-dynamic measurements could clarify this subject.
Typing errors should be eliminated.
The conclusion should be adapted according the revision.
Author Response
We appreciate you taking the time to offer us your comments and insights related to the paper. We found your feedback very constructive. We tried to be responsive to your concerns. We hope you find these revisions rise to your expectations
78-86: Liposomes consisting of EggPC are prepared by conventional, small scale technique. Although this composition is applicable, it is not explained why only this composition was selected, especially for potential human application.
→ This work is continuous research from Ref. 19. We will characterize the membrane lytic activities of the melittin inspired synthetic peptides toward various lipid compositions. We have included a brief description in the introduction section (p. 2, lines 52-56)
150-156: The design of retro analogs, with all their known limitations, should be explained in more detail, explicitly the design of the melittin analogs and the comparison to still published .analogs It would significantly improve the quality of this paper, when not only references are cited, but also the models and the different results would be discussed.
→ We have revised the text (p. 4, lines 166-170) to explain the meaning of the design and characterization of the retro analogs.
163-182: the calcein release assay was performed to compare the different lytic activities of the added peptide variants in solution. In both figures (2A and 2B) the same legend is used
→ We have reflected your comment and prepared new Figure 2(B).
Furthermore, figure 2B and the corresponding text is not comprehensible and need to be thoroughly checked.
→ We have revised the text (p. 5, lines 195-198) to explain the results of Figure 2(B).
227-231:The statement of proportional calcein release is not obvious. In addition, the legend of the x-axis of figure 5 needs to be revised
→ We have rewritten the description of the results from Figure 5 (p. 7, lines 253-257) to be more in line with your comments. The figure legend of the x-axis have been revised to “rLPE-St Contents”.
Figure 6: X-axis is not marked.
→ The figure legend of the x-axis have been revised to “pH”.
264-270: Size variation is seen under both conditions (S4), but more dominant at pH 5.0. Maybe, supporting influences caused this behavior. Comparative, fluid-dynamic measurements could clarify this subject.
→ You have raised an important point, however, we could not carry out fluid-dynamic measurements. Instead, we have added new Ref. 31 touching on the turbulence of vesicle surfaces observed in liposomal binding and fusion (p. 8, lines 294-295).
Typing errors should be eliminated.
→ This error has been corrected in accordance with the reviewer's comment.
The conclusion should be adapted according the revision.
→ We have revised the conclusion section (p. 9, lines 332-334).
Round 2
Reviewer 2 Report
Dear Authors,
The authors substantially revised the manuscript. The questions are well answered and additional explanations significantly improved the overall quality.
Thus, in conclusion I can recommend publishing of this manuscript
Best regards
Karola Vorauer-Uhl